# Regulatory Mechanisms of Retinal Photoreceptors Development at Single Cell Resolution

**DOI:** 10.3390/ijms22168357

**Published:** 2021-08-04

**Authors:** Meng Zhao, Guang-Hua Peng

**Affiliations:** 1Laboratory of Visual Cell Differentiation and Regulation, Basic Medical College, Zhengzhou University, Zhengzhou 450001, China; zhaomeng@gs.zzu.edu.cn; 2Department of Pathophysiology, Basic Medical College, Zhengzhou University, Zhengzhou 450001, China

**Keywords:** retinal photoreceptor, cone, rod, ipRGCs, development, regulation, single-cell

## Abstract

Photoreceptors are critical components of the retina and play a role in the first step of the conversion of light to electric signals. With the discovery of the intrinsically photosensitive retinal ganglion cells, which regulate non-image-forming visual processes, our knowledge of the photosensitive cell family in the retina has deepened. Photoreceptor development is regulated by specific genes and proteins and involves a series of molecular processes including DNA transcription, post-transcriptional modification, protein translation, and post-translational modification. Single-cell sequencing is a promising technology for the study of photoreceptor development. This review presents an overview of the types of human photoreceptors, summarizes recent discoveries in the regulatory mechanisms underlying their development at single-cell resolution, and outlines the prospects in this field.

## 1. Introduction

The human retina is located in the innermost layer of the ocular wall and is tightly attached to the choroid. It is responsible for light sensitive imaging. Visual information is transmitted to the visual center through nerve impulses on the retina along the visual path to form vision. The retina is composed of five major types of neurons: photoreceptors (PCs), bipolar cells (BCs), horizontal cells (HCs), amacrine cells (ACs), and retinal ganglion cells (RGCs). They are arranged in five layers: outer nuclear layer (ONL), outer plexiform layer (OPL), inner nuclear layer (INL), inner plexiform layer (IPL), and ganglion cell layer (GCL) [1]. 

Retinal photoreceptors convert light signals into neural signals, which is considered the first step in retinal phototransduction. Traditionally, retinal photoreceptors are distributed over the OPL and ONL. There are two subtypes, i.e., rods with thin rod-shaped outer segments and cones with tapered outer segments. There are approximately 120 million rods in the human retina, and they are sensitive to low-intensity light. The photosensitive pigment in rods is rhodopsin. Retinal cones are sensitive to stronger light and color and express three types of opsins: short wavelength-sensitive (blue), middle wavelength-sensitive (green), and long wavelength-sensitive (red) [2]. There are approximately 6 million cones in the human retina [3]. Rods and cones consist of five functional parts: outer segment, cilia, inner segment, nucleus, and synaptic body. Intrinsically photosensitive retinal ganglion cells (ipRGCs) are the last photosensitive cells to be discovered. They are responsible for non-image-forming vision, circadian entrainment, pupillary light reflex, and other non-image-forming photic responses [4]. They are related to mood, sleep, circadian rhythm, and study [5]. Their photosensitive substance is melanopsin [6]. Although ipRGCs and retinal rods and cones differ in shape and position, ipRGCs have been considered a third class of retinal photoreceptors in the past decades. There are six subtypes of ipRGCs (M1, M2, M3, M4, M5, and M6), all of which are located in the GCL accounting for approximately 4% of ganglion cells [7,8].

While three retinal photoreceptors differ in developmental time and location, they all develop from retinal progenitor cells (RPCs). The development and differentiation of retinal photoreceptors are regulated by various genes and proteins via five molecular processes: DNA replication, transcription, post-transcriptional modification, translation, and post-translational modification [9]. The misexpression of key genes can lead to serious photoreceptor diseases such as retinitis pigmentosa, Leber congenital amaurosis, Stargardt disease, Best disease, achromatopsia, cone dystrophies, and cone-rod dystrophy [10,11,12,13]. Additionally, abnormal epigenetic regulation can affect the developmental process and lead to developmental disorders [14]. To prevent and treat these inherited diseases, it is necessary and urgent to clarify the mechanism of retinal photoreceptor development.

Although the traditional RNA transcriptomics is often used to study the molecular mechanism of photoreceptor development at different stages in retinal development, the gene expression of single cell types cannot be accurately detected because of the cellular heterogeneity of retina [15]. With the rapid development of single-cell technologies, the development of each cell can be clearly observed, thus the molecule changes and regulation mechanism of retinal photoreceptor development can be detected more accurately. Single-cell transcriptomics has become the most promising tool for retinal photoreceptor developmental research [16]. This review focuses on the developmental regulatory mechanisms of photoreceptors at single cell resolution in the last 10 years and outlines the prospects of in the field.

## 2. Single-Cell Sequencing Analysis in Mice Retinal Photoreceptor Development

Previous studies found that the RGCs appear firstly at Embryonic day (ED) 8, HCs, ACs and cones appear at ED14, BCs, Müller glia cells and rods appear at Postnatal day (PD) 0 in mice [17]. According to the retinal time-point and known retinal cells markers (Table 1) [18,19], researchers tried to find more cell-specific markers and make clear the mechanisms of retinal photoreceptor development with single-cell sequencing.

Buenaventura et al. [20] used transgenic mice to overexpress LIM homeobox protein 4 (LHX4) which was enriched in cones during early retinal development and observed the genesis of cone cells and the gene changes during this process at ED14.5. The researchers also used two other cone markers, retinoic acid receptor RXR-gamma (*Rxrg)* and orthodenticle homeobox 2 (*Otx2*), to mark the LHX4^+^ cone cells and found that LHX4 is a specific marker in early developing cones and adult BCs. The single-cell transcriptome analysis classified retinal cells to nine clusters: three cone clusters, one neurogenic RPCs cluster, two multi-potent RPCs clusters, one HCs/ACs cluster, one RGC-AC RPC/Precursors cluster, and one RGCs cluster. Among three cone clusters, the second cone cluster specifically overexpressed *SLC7A3* and *SLC7A5,* both of which are thyroid hormone transporter genes. The three cone clusters all express thyroid hormone genes, such as thyroid hormone receptor beta (*Thrb),* which was specific to cones [20]. Although previous studies have illustrated that the thyroid hormone play an important role in M-cone differentiation, the exact influence of these cone-specific thyroid hormone genes in cone development needs more exploration [21]. Buenaventura et al. compared his dataset with other datasets and found that many cone genes are dysregulated in the mutant of neural retina leucine zipper (*Nrl*) which direct rod formation, while some genes found in early cones, such as *Rxrg*, are not dysregulated in early rods, but are only significantly changed in mature rods [22,23].

Giudice et al. [24] obtained single retinal cells from ED15.5 retina of mice to analyze the process of retinal development, 14 clusters cells were identified according to their known gene markers with t-distributed stochastic neighbor embedding (t-SNE) analysis. Among them, cluster 6 was identified as photoreceptors that are positive for *Otx2*, cone-rod homeobox (*Crx),* and *Thrb*. The analysis with branched expression analysis modeling (BEAM), which were used to trace back the genes related with the temporal transitions during differentiation found that castor zinc finger 1 (*Casz1*)*, Thrb* and Meis homeobox 2 (*Meis2*) were enriched in cones. RNA-velocity analysis, which correlated the transcriptional process with cell cycle phases (G1, G1/S, S/G2, G2/M, M) found that transcription factors (TFs) PR domain containing 1, with ZNF domain (*Prdm1*), *Otx2*, and neurogenic differentiation 1 (*Neurod1*), which directed cells fate to photoreceptors, were enriched in the G2/M group of cells, which indicated that *Prdm1*, *Otx2*, and *Neurod1* positive cells were in a proliferative phase. The researchers also showed that during the process of cell fate specification, metabolic genes (*Ldhb* in AC/ HCs and RGCs, *Ldha* in photoreceptors) were enriched.

A study containing multiple time points (10 stages from ED11 to PD14 which covered the whole process of retina development) with the Droplet-Based single cell RNA-sequencing made the development of the photoreceptor clearer [25]. Retinal cones which arise from early RPCs appear at ED14, and retinal rods which arise from *Otx2*-positive progenitors appear at the early postnatal stage. Neurogenic RPCs expressed bHLH factors such as neurogenin 2 (*Neurog2*)*,* atonal bHLH transcription factor 7 *(Atoh7*)*,* and oligodendrocyte transcription factor 2 (*Olig2*), which could direct neuronal fate. Early neurogenic RPCs expressed TF genes that regulate specification of early-born cell types such as RGCs, horizontal cells, and GABAergic amacrine cells. Similarly, late-neurogenic RPCs express TF genes controlling late-born cell types. In this paper, researchers found a novel transcriptional factor nuclear factor I (NFI) which could promote the differentiation of BCs and Müller cells by activating the regulator genes, while restraining the regulator genes that control the genesis of rods. Mutation of *NFI* would reduce the TFs which maintain RPCs state and late-cell determination such as paired box 6 (*Pax6*), retina, and anterior neural fold homeobox (*Rax*), and visual system homeobox 2 (*Vsx2*) and the Notch signaling pathway. These observations indicate that *Nfi* regulates the genes related with cell proliferation during retinal development. Deletion of *Notch1* would lead to the exit of cell cycle and overproduction of rod photoreceptors in advance with the down-regulation of cell cycle genes such as fibroblast growth factor 15 (*Fgf15*)*,* cell division cycle 20 (*Cdc20*)*,* crystallin, mu (*Crym*), and rod regulators such as *NeuroD1*, *Math3*, retinol binding protein 3, interstitial (*Rbp3*), *Blimp1*, which illustrate that Notch signaling can control retinal cell fate especially inhibiting the precocity of photoreceptor cells by regulating cell cycle exit [26].

## 3. Single-Cell Sequencing Analysis in Human Retinal Photoreceptor Developmental Process

### 3.1. Human Fetal Retina Development

Human eyes develop from 3.5 weeks after conception (PCW) to 5 months after birth [27]. Mellough et al. [28] collected human retina samples from PCW 4.6–18 (three developmental windows) and human adult retina, identified different classes of retina cells and analyzed the regulatory network of non-coding RNAs and mRNA during retina development with RNA transcriptomics. They found that the up-regulation of genes such as eyes shut homolog (*EYS*) and *PRDM1*, which were required for the correct function and survival of photoreceptors, the up-regulation of TFs that activated rod development, as well as genes that were involved in photo-transduction in rods, suggested the initiation of rod emergence and function during PCW 12–18.

Although this research explained the developmental process of early human retina, it could not distinguish the exact retina cells and the relationship between RNA expression and single cell type because of the cellular heterogeneity of retina. Hu et al. [29] gathered human fetal retina and RPE at weeks 5–24 (10 developmental stages), identified 21 cluster cells according to gene markers (Table 2) such as RPEs, RPCs, HCs, ACs, BCs, photoreceptors, Müller glia cells, microglia cells, showed the order of neural genesis, and analyzed the dynamic transcriptional change with single-cell RNA sequencing. RPCs (peak at Week 9) and RGCs (peak at Week 9) appeared at Week 5, HCs developed at Week 5 (peak at Week 9), ACs were observed at Week 8 (peak at Week 17), PCs were observed at Week 9 (peak at Week 17), Müller glia cells and BCs were observed at Week 13. With a method named single-cell regulatory network inference and clustering (SCENIC), they found RPCs were *SOX2^+^* and *NEUROD1^−^* at weeks 9 and 13. At Week 13, the *NEUROD1^+^SOX2^−^* cells adjacent to the RPE were photoreceptors, which expressed *NEUROD1, NRL, CRX, RAX2*, and *PRDM1*. At Weeks 23 and 24, the photoreceptors became more matured, which also expressed genes related to photo-transduction, such as rhodopsin (*RHO*), S-antigen visual arrestin (*SAG*), G protein subunit gamma transducin 1 (*GNGT1*), and cyclic nucleotide gated channel subunit alpha 1 (*CNGA1*). They also found the interaction between two neuro-trophic factors midkine (MDK) and pleiotrophin (PTN) in RPEs and that receptor protein tyrosine phosphatase type Z (*PTPRZ1*) in photoreceptors was essential for the development of both RPEs and photoreceptors [29].

### 3.2. Human Retinal Organoids Development

In view of the difficulties and ethical requirements of human retina acquisition during development, the genesis of retinal organoids has been gradually used by researchers to simulate the development of retina. Retinal organoids are mainly generated from human induced pluripotent stem cells (hiPSCs) and human embryonic stem cells (hESCs) [30]. One such research focused on the differentiation of RPC into retinal neurons at the early stage of retinal development: Day 25 to Day 35 using hESCs-derived retinal organoids [31]. Cells in the early retina neurogenesis stage could be classified into three groups: neurons (markers: *ATOH7, PRDM1, ONECUT1*, and *PROX1*), neurogenic RPCs that can commit to neurons, and multi-potent RPCs (markers: *VSX2* and *PAX6*, related to glycolysis and lipid synthesis). Additionally, it was found that the transition of multipotent RPCs to neurons began on Day 28, when achaete-scute family bHLH transcription factor 1 (*ASCL1*), a key molecule directing this process, was expressed. To further explore regulators of retinal neuron commitment, genes that were co-expressed with *ASCL1* were investigated and some top candidates were found: NEUROG 2 (*NGN2*)*,* INSM transcriptional repressor 1 (*INSM1*), basic helix-loop-helix family member e22 (*BHLHE22*)*,* immunoglobulin superfamily containing leucine rich repeat 2 (*ISLR2*), and cyclin D1 (*CCND1*). Among these, CCND1, which was a cell cycle-related protein, promotes *ASCL1* expression and RPC development in a cell cycle-independent way. Clustering of genes that exhibited dynamic expression changes during early retina neurogenesis revealed four clusters. Those genes in the first cluster began to be up-regulated as the retinal neurons form and were associated with neuronal differentiation as revealed by Gene Ontology analysis. Genes in the second cluster were constitutively expressed by RPCs and transiently up-regulated on Day 28 and were associated with changes in chromosome and DNA state. Genes in the third and fourth clusters began to be up-regulated in the neurogenic stage and were associated with self-renewal of RPCs. The genes in the second cluster, including forkhead box N4 (*FOXN4*), *LHX4*, nuclear receptor subfamily 2 group E member 3 (*NR2E3*), and hes family bHLH transcription factor 1 (*HES1*), may play important roles in directing RPC development. Further, gene set enrichment analysis revealed that the Notch and Wnt signaling pathways were most strongly related with RPC fate commitment in early retinal development, and the chromatin state regulators high mobility group AT-hook 1 (HMGA1), bromodomain adjacent to zinc finger domain 2B (BAZ2B), and MDS1 and EVI1 complex locus (MECOM) were identified, indicating that chromatin remodeling was involved in early retina genesis, which remained to be verified [31].

The fovea comprises two layers of pigment epithelial cells and retinal cones and is present uniquely in primates [32]. In the fovea, cones and bipolar cells are in one-to-one contact, allowing highly sensitive and accurate vision, which is termed central vision [33]. The fovea inner plexiform layer and inner nuclear layers start to develop at fetal week 25 [34]. Retinal cone numbers increase with increasing thickness of the ONL and OPL after birth. The inner and outer segment lengths match those of peripheral cones at 15 months and increase 4 times by 13 years of age [35]. The retinoic acid signaling pathway is essential for human fovea formation [36].

To study the gene regulation process in macular cones, retinal organoids derived from human multi-potential stem cells are required [37,38]. One study using hESC-derived retinas to mimic the human macular area at Day 15 and 1, 3, 6.5, and 9 months, revealed that *VSX2^+^* RPCs are committed to cone photoreceptors and the cone photoreceptor-specific marker, recoverin (*RCVRN*), appeared on Day 105 [39]. Rod and cone markers were expressed as of 3 months up to 9 months. The expression levels of opsin 1, medium wave sensitive (OPN1MW) and opsin 1, long wave sensitive (OPN1LW) were higher than that of opsin 1, short wave sensitive (OPN1SW), which was detected at 3 months. Gene Ontology analysis of a pool of retinal cells comprising cones, rods, and Müller glia at 8 months-old revealed that most of the retinal cells were rods and cones, with a cones-to-rods ratio of 1.4:1, which was similar to that in the human macula. Cone and rod genes function in visual perception, sensory perception of light stimulus, and photo-transduction. The authors also identified an electrophysiological functionality of cones [39].

### 3.3. Comparisons between Human Fetal Retina and Retinal Organoids Development

The developmental order and transcriptional differences between human fetal retina and retinal organoids were compared by researchers [40]. Development at Day 60 of retinal organoids was similar to that at fetal day (FD) 59 of human fetal retina when predominant cell types were retinal progenitors and retinal ganglion cells, whereas small numbers of amacrine cells, horizontal cells, and photoreceptors were also detected. Further, the researchers identified a transition cell population 1 (T1, marker: *ATOH7*) that did not express progenitor markers or specific neuronal markers, but highly expressed *ATOH7* and Notch components and resembled the neurogenic RPCs in mouse and hESC-derived organoids. However, delta-like canonical Notch ligand 3 (*DLL3*) expression was lower in fetal retina than in retinal organoids. Development at Day 104 of retinal organoids was similar to that at FD82 of human fetal retina when progenitor cells and BCs were the predominant cell types, and ACs, HCs, photoreceptors, and Müller glia cells were also detected. Two other clusters of transition cells were found in this stage: T2 cells, which expressed *PRDM13* and promoted the differentiation of amacrine cells, and T3 cells, which highly expressed *FABP7* and promoted the differentiation of photoreceptors and bipolar cells. Photoreceptors began to expand quickly from Day 90, whereas the number of T1 cells began to decrease and T3 cells were maintained in retinal organoid. Development at Day 205 of retinal organoids was similar to that at FD125 of human fetal retina when photoreceptors were the major cell type, followed by BCs and ACs, HCs, and RGCs. Neurofilament medium (NEFM), which was highly expressed during every stage of fetal retinal development, was much less expressed in retinal organoids, whereas glutamate decarboxylase 1 (GAD1) and transcription factor AP-2 alpha (TFAP2A), which are amacrine-specific proteins, were highly expressed during all stages of retinal organoid development. In retinal organoids, the structure of the inner retina was disrupted at Day 205, revealing a limitation of retinal organoid and suggesting that retinal development may be regulated by adjacent cells [40].

Wang et al. [41] cultured retinal organoids from hESCs (9 different cluster retinal cells from 5 time points were identified), built associations between different cell clusters and focused on the pseudo time molecular regulation of retinal progenitor cells, photoreceptors, retinal pigment epithelium and ganglion cells. Photoreceptor precursors appeared at Day 36 and increased with time, the cone photoreceptor appeared substantially at Day 186, rod photoreceptor cells appeared at Days 126 and 186, RGCs appeared in quantities early at Day 36, and declined from that time. They found that some pathways were enriched during photoreceptor development such as PI3k-Akt signaling, retinal metabolism, photo-transduction, and dopaminergic neurogenesis. In addition, they discovered that Insulin receptor (INSR), which was a tyrosine kinase receptor and was mainly expressed in photoreceptor was a key regulator during photoreceptor maturation. It could establish interactions with ligand calmodulin 2(CALM2) in RPCs and RGCs, which in turn could influence the development of photoreceptors. In identification of ganglion cells, a cluster of ganglion cells expressed ipRGCs marker peripherin (PRPH) and gene RAR related orphan receptor B (RORB), which is associated with retinoid metabolism photoreceptor and photoreceptor maturation markers, which may indicate the existence of human ipRGCs [41].

## 4. Comparisons between Mice Retina and Retinal Organoids Development

The developmental differences between hiPSC-derived retinal organoids and mice were made by Lu et al. [42]. The period during which early-stage RPCs transition to late-stage RPCs in mice and retinal organoids was established to be ED16-ED18 and PCW 11–15 weeks, respectively. The researchers found differences in the expression of some genes throughout development and between the two species. Expression of *Olig2*, *Neurog2*, and BTG anti-proliferation factor 2 (*Btg2*), which were highly expressed during mouse neurogenic RPC development, increased only in the late stage in human neurogenic RPCs, whereas insulin related protein 2 (*Isl2*), which was expressed at a low level in mouse cones, was highly expressed in human cones, together with its cofactor LIM domain only 4 (*LMO4*)*. HKR1*, which was expressed specifically in human rods during retinal development, was not expressed during the development of mouse retina. *Atoh7*, which was highly expressed in the early stage of mouse neurogenic RPCs in this and other studies, was also detected in human photoreceptor/bipolar precursors and immature cones, regulates the ratio of human cones to rods. *ATOH7* was expressed at different levels in the central and peripheral retina, and was barely expressed after PCW10 and PCW22, respectively, which further illustrated that *ATOH7* promoted cone genesis in the human fetal retina. Differentially expressed genes between central and peripheral retinal cells at PCW 20 and PD8 included cytochrome P450 family 26 subfamily A member 1 (*CYP26A1*), iodothyronine deiodinase 2 (*DIO2)*, cyclin dependent kinase inhibitor 1A (*CDKN1A*), annexin A2 (*ANXA2*), and *FR2B*. *CYP26A1* was specifically expressed in RPCs and Müller glia cells in the primate fovea, suggesting *CYP26A1* as a candidate regulatory molecule in the macula. Another interesting potential regulator of macular development is cellular communication network factor 2 (CTGF), which is highly expressed in Müller glia cells and is a downstream molecule of the Hippo pathway. CTGF is activated by FGF15 or its homolog fibroblast growth factor 19 (FGF19), which is expressed in early RPCs and can degrade retinoic acid to confer macular identity. The cones in hiPSC-derived retinal organoids had some specific features. They expressed cone-specific genes, including *THRB*, *ISL2*, *LMO4*, spalt like transcription factor 3 (*SALL3*), and VPS29 retromer complex component (*DC7*), which exhibited low expression of the precursor-enriched transcription factor CRX. They also expressed some neurogenic factors that are not specifically related to photoreceptors, such as LIM homeobox 9 (LHX9), nescient helix-loop-helix 1 (NHLH1), and SRY-box transcription factor 11 (SOX11) [42].

## 5. Discussion

The retina is extremely important for human life. Photoreceptors are the initiators of the visual process and therefore deserve to be studied thoroughly. Normal and organized photoreceptor development is crucial for their own functionality as well as that of other retinal neurons. Therefore, elucidating the order and regulatory mechanism of photoreceptor development will aid in preventing and curing related diseases. Three types of retinal photoreceptors have been discovered to date. Rods and cones are structurally and functionally well characterized, and their development has been extensively studied (Figure 1 and Figure 2). In contrast, the function and subtypes of ipRGCs are still under investigation.

Due to the heterogeneity of retinal cells, single cell sequencing is very important for the study of retinal development. By using a variety of cell markers of the retina with single-cell analysis, the classification of cells in the process of retinal photoreceptor development is more detailed. It is easier to find new and more specific photoreceptor markers in different periods, for example, three cone clusters were identified and *Lhx4* was identified as a specific marker of early cones in the study by Buenaventura et al. [20]. Moreover, the differentiation process of photoreceptors from which RPCs clusters become more explicit was examined in the research of Giudice et al. [24]. It is well known that transcription factors regulate the fate of cells, so finding transcription factors that regulate the development of photoreceptor cells is very important for understanding the developmental process of photoreceptor cells such as NRL, which directs rod fate, and a novel Atoh7, which directs cone fate.

Due to the differences in the gene homology between mouse retina and human retina, and the lack of fovea in mouse retina, retinal organoids have advantages in studying the development of photoreceptor cells. However, previous studies have found that the morphology of retinal organs will be damaged in the later stage, which indicate the essential effect of adjacent cells to retina. Thus, how to cultivate a more suitable environment for the growth of retinal organs seems to be a challenge. The roles of epigenetics regulatory mechanisms in photoreceptor development are poorly known. The regeneration of photoreceptors from stem cells may be blocked by epigenetic factors, which remains to be investigated. In addition, although some researchers compare the transcriptional differences between different datasets, the differences of sample type and number may lead to false positive or statistically significant decrease.

In the future, the joint application of multiple omics technologies will be a promising direction. In addition, single-cell RNA sequencing and spatiotemporal transcriptomics mapping are emerging, which are promising technologies to study retinal development. In short, single-cell RNA sequencing has substantially revolutionized our understanding of photoreceptor development. Despite the advantages of single-cell RNA sequencing, we still need improved analytical methods and tools to mine single-cell sequencing data and explore regulatory networks including DNA, RNA, and proteins. Additionally, newly discovered molecules functioning during photoreceptor development need to be experimentally verified. With continuous development in technologies and innovative solutions, the exact mechanism of photoreceptor development may be unraveled in the future.

## Figures and Tables

**Figure 1 ijms-22-08357-f001:**
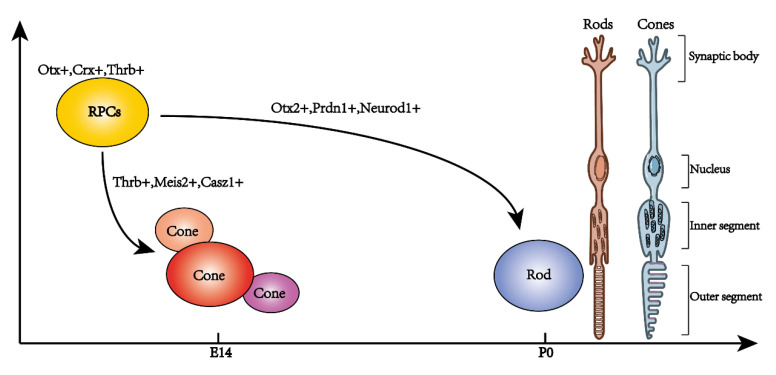
The developmental process of retinal photoreceptors in mice. RPCs: retinal progenitor cells, Otx^+^, Crx^+^, and Thrb^+^, tended to differentiate into photoreceptors. Three clusters of retinal cones, Thrb^+^, Meis^+^, and Casz1^+^, were discovered at embryonic 14 days. Retinal rods appeared after birth.

**Figure 2 ijms-22-08357-f002:**
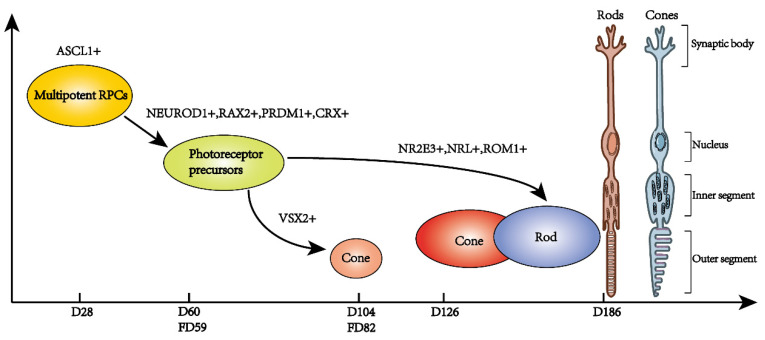
The developmental process of retinal photoreceptor in human fetal retina and retinal organoids. Multipotent RPCs of retinal organoid began to differentiate into neurons at approximately retinal organoid day 28 (D28) days when ASCL1 began to express. NEUROD1^+^, RAX^+^, PRDM1^+^, and CRX^+^ RPCs were differentiated into photoreceptor precursors at about retinal organoid day 60 (D60) which was similar to fetal day 59 (FD59). VSX2^+^ photoreceptor precursors differentiate into cones at D104, which was similar to fetal day 59 (FD82). NR2E3^+^, NRL^+^ and ROM1^+^ photoreceptor precursors differentiate into rods at D186.

**Table 1 ijms-22-08357-t001:** Common gene markers of main retinal cells in mice.

Retinal Subtypes		Gene Markers
PCs	Cones	Rxrg, Otx2, Lhx4 *, Gngt2, Gnb3, Opn1sw, FABP7 **
	Rods	Blimp1, Crx, Otx2, Nrl, Math3, Rbp3, Rax, Epha8, Neurod1, Nr2e3, Rom1, Rbp3, Rhodopsin **, ABCR **, Recoverin **
ACs		Tfap2a *, Tfap2b *, Gad1, Glyt1, Onecut2, Prox1, Dlx1, Pax6, Pcdh17, Pou3f1
BCs		Og9x, Lhx3, Car8, Car10, Nfasc, Otx2 **, Lhx4
HCs		Onecut1, Lhx1, Onecut2, Prox1
RGCs		Sox11, Atoh7, NF68, Ebf3, Isl1, Pou4f2, Pou6f2, Elavl4, Pou4f1, Islr2, Syt4, Ebf1/3, L1cam, Brn3b
Müller glia		Oaz1, Pebp1, Apoe, clusterin, Sox2, μ-crystallin, Dkk3
RPCs	Primary RPCs	Sox2, Fos, Hes1, Pax6, Vsx2, Lhx2, Ccnd1, Cdk4
	Neurogenic RPCs	Atoh7, Olig2, Neurog2, Sox11, Onecut1/2, Dlx1/2, Prdm1, Otx2, Ascl1, Hes6

* genes specific to immature cells. ** genes specific to mature cells.

**Table 2 ijms-22-08357-t002:** Common gene markers of main retinal cells in human.

Retinal Subtypes		Gene Markers
PCs	Cones	ARR3, GUCA1C, PDE6C, PDE6H, ISL2, ATOH7, THRB, LMO4, SALL3, DC7, CRX, OPN1LW, OPN1SW, OPN1MW
	Rods	GNGT1, CNGB1, GNAT1, NRL, NR2E3, PDE6A, RHO, PDE6B, HKR1, CRX, ROM1, SLC24A1
ACs		MEIS2, GAD1, GAD2, TFAP2A, PAX6, SLC6A1, ATP1B1
BCs		VSX1, VSX2, CHN2, SCN3A, TRPM1, LRTM1, TMEM215, PLXDC1, NETO1, CA10, ST18, SLC4A10, KCNMA1, ISL1, GRM6, OTX2
HCs		ONECUT1, ONECUT2, ONECUT3, NTRK1, SEPT4, TPM3, NDRG1, SEPT7
RGCs		HUC/D, GAP43, SNCG, RBPMS NEFL, TUBB3, POU4F1, SLC17A6
RPEs		SERPINF1, TYR, MITF, RPE65, BEST1, TTR, PMEL
Müller glia		PLP1, TF, SOX2, GFAP, S100B, SLC1A3, RLBP1, HES1, GLUL, CLU
Microglia cells		CX3CR1, C1QA, C1QB, C1QC
RPCs	Primary RPCs	SFRP2, MKI67, SOX2, HES5, FZD5, ASCL1, PAX6, RAX, NESTIN, TOP2A
	Neurogenic RPCs	ATOH7, PRDM1, ONECUT1, PROX1, VSX2

## Data Availability

Not applicable.

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
