# Peer review of "Regulatory Mechanisms of Retinal Photoreceptors Development at Single Cell Resolution"

_ijms, 2021, doi:10.3390/ijms22168357_

Round 1

Reviewer 1 Report

In this manuscript, the authors describe and compare the single-cell sequencing analysis photoreceptors development including the melanopsin retinal ganglion cells, and the development of the retinal organoid. The argument is very interesting, and the structure makes the manuscript easy to read and understand. The figures help understand the manuscript.

However, the grammar and sentence construction in the introduction needs to be improved because under current conditions they make it difficult for the reader to understand.

In addition, the authors should revise the definition of abbreviation throughout the text.

For example:

Line 66-67: The authors should define what “E” and “P” mean.  “Previous studies found that the RGCs appear firstly at E8, HCs, ACs and cones appear at E14, BCs, 66 Müller cells and rods appear at P0 in mice [17].

Line 70: The authors should define “LIM”. “Buenaventura et.al used transgenic mice to overexpress LIM homeobox protein…”

Author Response

Dear Reviewer:

Thank you for your valuable comments. We have carried out detailed modification and inspection according to your suggestions.

Below is the reply one by one:

Point 1: the grammar and sentence construction in the introduction needs to be improved because under current conditions they make it difficult for the reader to understand.

Response 1: We have invited a native English-speaking colleague to polish our English grammar and sentence structure, and the revised version is in the newly submitted manuscript.

Point 2: the authors should revise the definition of abbreviation throughout the text.

For example:

Line 66-67: The authors should define what “E” and “P” mean.  “Previous studies found that the RGCs appear firstly at E8, HCs, ACs and cones appear at E14, BCs, 66 Müller cells and rods appear at P0 in mice [17].”

Line 70: The authors should define “LIM”. “Buenaventura et.al used transgenic mice to overexpress LIM homeobox protein…”

Response 2: We have checked the abbreviations throughout the text and make sure that the abbreviations have been defined and the gene symbols have been explained with official full name when they first appeared.

For example:

  1. The line 66-67 has been corrected to “Previous studies found that the RGCs appear firstly at E8Embryonic day (ED) 8, HCs, ACs and cones appear at ED14, BCs, Müller glia cells and rods appear at P0Postnatal day (PD) 0 in mice.”
  2. The complete sentence of line 70 is: “Buenaventura et.al used transgenic mice to overexpress LIM homeobox protein 4 (Lhx4) which was enriched in cones during early retinal development to observe the genesis of cone cells and the gene changes during this process at E14.5.” In this sentence, the LIM homeobox protein 4 is the official full name of Lhx4 in the NCBI database.

In order to make the abbreviations clearer, we made an abbreviation table attached to the article, as follows:

Abbreviations

PCs

photoreceptors

BCs

bipolar cells

HCs

horizontal cells

ACs

amacrine cells

RGCs

retinal ganglion cells

ONL

outer nuclear layer

OPL

outer plexiform layer

INL

inner nuclear layer

IPL

inner plexiform layer

GCL

ganglion cell layer

ipRGCs

Intrinsically photosensitive retinal ganglion cells

RPCs

retinal progenitor cells

ED

Embryonic day

PD

Postnatal day

LHX4

LIM homeobox protein 4

Rxrg

retinoic acid receptor RXR-gamma

Otx2

orthodenticle homeobox 2

SLC7A3

solute carrier family 7 member 3

SLC7A5

solute carrier family 7 member 5

Thrb

thyroid hormone receptor beta

Nrl

neural retina leucine zipper

t-SNE

t-distributed stochastic neighbor embedding

Crx

cone-rod homeobox

BEAM

branched expression analysis modeling

Casz1

castor zinc finger 1

Meis2

Meis homeobox 2

TFs

transcription factors

Prdm1

PR domain containing 1, with ZNF domain

Neurod1

neurogenic differentiation 1

Ldhb

lactate dehydrogenase B

Ldha

lactate dehydrogenase A

Neurog2

neurogenin 2

Atoh7

atonal bHLH transcription factor 7

Olig2

oligodendrocyte transcription factor 2

NFI

nuclear factor I

Pax6

paired box 6

Rax

retina and anterior neural fold homeobox

Vsx2

visual system homeobox 2

Notch1

notch receptor 1

Fgf15

fibroblast growth factor 15

Cdc20

cell division cycle 20

Crym

crystallin, mu

Rbp3

retinol binding protein 3, interstitial

PCW

weeks after conception

EYS

eyes shut homolog

SCENIC

single-cell regulatory network inference and clustering

RHO

rhodopsin

SAG

S-antigen visual arrestin

GNGT1

G protein subunit gamma transducin 1

CNGA1

cyclic nucleotide gated channel subunit alpha 1

MDK

midkine

PTN

pleiotrophin

PTPRZ1

Receptor protein tyrosine phosphatase type Z

hiPSCs

human induced pluripotent stem cells

hESCs

human embryonic stem cells

ASCL1

achaete-scute family bHLH transcription factor 1

NGN2

NEUROG 2

INSM1

INSM transcriptional repressor 1

BHLHE22

basic helix-loop-helix family member e22

ISLR2

immunoglobulin superfamily containing leucine rich repeat 2

CCND1

cyclin D1

FOXN4

forkhead box N4

NR2E3

nuclear receptor subfamily 2 group E member 3

HES1

hes family bHLH transcription factor 1

HMGA1

high mobility group AT-hook 1

BAZ2B

bromodomain adjacent to zinc finger domain 2B

MECOM

MDS1 and EVI1 complex locus

RCVRN

recoverin

OPN1MW

opsin 1, medium wave sensitive

OPN1LW

opsin 1, long wave sensitive

OPN1SW

opsin 1, short wave sensitive

FD

fetal day

DLL3

delta like canonical Notch ligand 3

NEFM

Neurofilament medium

GAD1

glutamate decarboxylase 1

TFAP2A

transcription factor AP-2 alpha

INSR

Insulin receptor

CALM2

calmodulin 2

PRPH

peripherin

Btg2

BTG anti-proliferation factor 2

Isl2

insulin related protein 2

LMO4

LIM domain only 4

CYP26A1

cytochrome P450 family 26 subfamily A member 1

DIO2

iodothyronine deiodinase 2

CDKN1A

cyclin dependent kinase inhibitor 1A

ANXA2

annexin A2

CTGF

cellular communication network factor 2

FGF19

fibroblast growth factor 19

SALL3

spalt like transcription factor 3

DC7

VPS29 retromer complex component

LHX9

LIM homeobox 9

NHLH1

nescient helix-loop-helix 1

At last, thank you again for your care and patience, and wish you a good health.

Reviewer 2 Report

In this manuscript, overview summarizes studies of photoreceptors development at single-cell resolution. 

In my opinion, the title of the article is not very good. It might be worth giving a more detailed title, for example, by adding the words "regulatory mechanisms" at the beginning or like

Authors need to refer to publications  after each statement: line 70 (after et.al.), line 74 (after BC), line 86 (after et.al.), line 118 (after birth), line 148 (after organoids), line 154 (after CCND1), line 159 (after analysis), line 169 (after primates), line 86 (after week 25), line 211 (after et.al.), line 228 (after et.al.), line 266 (after study). It is imperative to cite primary sources wherever appropriate.

Add «et.al.» after family at line 82, 267.

Author Response

Dear Reviewer:

Thank you for your valuable comments. We have carried out detailed modification and inspection according to your suggestions.

Below is the reply one by one:

Point 1: In my opinion, the title of the article is not very good. It might be worth giving a more detailed title, for example, by adding the words "regulatory mechanisms" at the beginning or like

Response 1: Thank you for your good suggestion. We have changed the title to “Regulatory mechanisms of retinal photoreceptors development at single cell resolution”.

Point 2: Authors need to refer to publications after each statement: line 70 (after et.al.), line 74 (after BC), line 86 (after et.al.), line 118 (after birth), line 148 (after organoids), line 154 (after CCND1), line 159 (after analysis), line 169 (after primates), line 86 (after week 25), line 211 (after et.al.), line 228 (after et.al.), line 266 (after study). It is imperative to cite primary sources wherever appropriate.

Response 2: We have inserted the references according to your suggestion and checked the full text to make sure there are no missing references.

  1. We have inserted references to line 118 (after birth), line 171 (after week 25), line 169 (after primates) and line 266 (after study).We have inserted the references after Buenaventura et.al on the line 70, the Giudice et.al on the line 86, the Wang et.al on the line 211 and the Lu et.al on the line 228 which lies in the first sentence of the passage discussing one study to make clearer and more appropriate.
  2. The three places on the line 74 (after BC), line 154 (after CCND1) and line 159 (after analysis) owned the same reference with the sentences before them respectively.

Point 3: Add «et.al.» after family at line 82, 267.

Response 2: We have added «et.al.» after the Buenaventura on line 82, Buenaventura on the line 266 and Giudice on the line 267.

At last, thank you again for your care and patience, and wish you a good health.